# Estimating the distribution of parameters in differential equations with repeated cross-sectional data

**Hyeontae Jo[1,2]◉, Sung Woong Cho[3]◉, Hyung Ju Hwang◉[4]***

**1** Department of Mathematics, Korea University Sejong Campus, Sejong, Republic of Korea, **2** Biomedical Mathematics Group, Pioneer Research Center for Mathematical and Computational Sciences, Institute for Basic Science, Daejeon, Republic of Korea, **3** Stochastic Analysis and Application Research Center, Korea Advanced Institute of Science and Technology, Daejeon, Republic of Korea, **4** Department of Mathematics & Graduate School of AI, Pohang University of Science and Technology, Pohang, Republic of Korea

◉ These authors contributed equally to this work.
* hjhwang@postech.ac.kr

**Data Availability Statement:** All relevant data are within the manuscript and its Supporting information files.

**Funding:** HJH was supported by the National Research Foundation of Korea(NRF) grant funded

## Abstract

Differential equations are pivotal in modeling and understanding the dynamics of various systems, as they offer insights into their future states through parameter estimation fitted to time series data. In fields such as economy, politics, and biology, the observation data points in the time series are often independently obtained (i.e., Repeated Cross-Sectional (RCS) data). RCS data showed that traditional methods for parameter estimation in differential equations, such as using mean values of RCS data over time, Gaussian Process-based trajectory generation, and Bayesian-based methods, have limitations in estimating the shape of parameter distributions, leading to a significant loss of data information. To address this issue, this study proposes a novel method called Estimation of Parameter Distribution (EPD) that provides accurate distribution of parameters without loss of data information. EPD operates in three main steps: generating synthetic time trajectories by randomly selecting observed values at each time point, estimating parameters of a differential equation that minimizes the discrepancy between these trajectories and the true solution of the equation, and selecting the parameters depending on the scale of discrepancy. We then evaluated the performance of EPD across several models, including exponential growth, logistic population models, and target cell-limited models with delayed virus production, thereby demonstrating the ability of the proposed method in capturing the shape of parameter distributions. Furthermore, we applied EPD to real-world datasets, capturing various shapes of parameter distributions over a normal distribution. These results address the heterogeneity within systems, marking a substantial progression in accurately modeling systems using RCS data. Therefore, EPD marks a significant advancement in accurately modeling systems with RCS data, realizing a deeper understanding of system dynamics and parameter variability.

by the Korea government(MSIT) (RS-2023-00219980 and RS-2022-00165268) and by the Ministry of Education (2023RIS-009) HJ was supported by the National Research Foundation of Korea (NRF) (RS-2024-00357912) and Korea University Grant (K2411031). SWC received a salary from National Research Foundation of Korea (NRF) grant funded by the Ministry of Science and ICT (KR) (RS-2024-00462755) and funded by the Korea government (MSIT) (RS-2019-NR040050). The funders had no role in study design, data collection and analysis, decision to publish, or preparation of the manuscript.ration of the manuscript.

**Competing interests:** The authors have declared that no competing interests exist.

## Author summary

Observation data points in biological experiments are often obtained from independent objects over time (i.e., Repeated Cross-Sectional (RCS) data). One method to obtain biological information from RCS data is to design a mathematical model and determine the range of parameters in the model that fits the data correctly. Here, the shape of parameter distributions provides data information, such as heterogeneity in biological phenomena. However, we found that classical estimation methods fail to determine the correct shape of parameter distributions, leading to a significant loss of data information. To address this, we propose a novel method, Estimating Parameter Distribution (EPD), to catch the accurate shape of parameter distributions originating from RCS data. Specifically, EPD involves three steps: First, we generated several artificial time series data from the RCS data. Second, we evaluated the suitability of these artificial data by fitting them to the mathematical model. Finally, we determined the best artificial data that matched the RCS data. The results showed that EPD provided more accurate estimations of distributions for the growth rate of Amyloid beta peptides and the half-saturation constant of the viral population compared to previous research results. Specifically, EPD revealed correct parameter distribution with heterogeneity in parameters.

## Introduction

Differential equations play a crucial role in modeling the evolution of various systems, offering scientific and mechanistic insights into the physical and biological phenomena and enabling predictions of their future states. These phenomena can be analyzed by parameters of the differential equation that fit its solutions to time series data. However, in fields such as economy, politics, or biology, data observations are often Repeated Cross-Sectional (RCS) (i.e., data is collected over time measuring the same variables with different samples or populations at each time point) [1–5]. For example, Sara, et al. analyzed the degree of tumor size suppression over time in rats with different types of drugs, using an exponential growth model [6]. As the mice died during the experiment, observation data from the experiment cannot be connected per unit time (i.e., RCS data). Jeong et al. utilized time series data on the PER protein levels in Drosophila to analyze neuron-dependent molecular properties [7]. However, measuring PER levels at each time point necessitated the sacrifice of the flies, resulting in limitations in the collection of RCS data. RCS data also includes regular surveys in society that collect the changing opinions of different individuals. Public polls by Gallup, the Michigan Survey of Consumers [8, 9], records of congressional roll calls [10], Supreme Court cases [11], and presidential public remarks [12] are all examples of RCS data.

Fitting the parameters with cross-sectional data or time-series data is feasible with classical optimization methods, yet handling RCS data can be challenging. While several methods have been used, their applicability is constrained. For example, one common method involves using the mean values at each time point for parameter estimation [7]. While this simplifies the analysis of RCS data, it significantly reduces the data information. To preserve the data information, Gaussian Process-based time series generation (GP) have been utilized for model calibration [13]. In particular, GP produces continuous-time trajectories through the mean and covariance of RCS data, enabling us to use traditional parameter estimation techniques. Nonetheless, because the GP method relies solely on the mean and covariance, the estimation results from GP-based algorithms tend to be unimodal [14–17]. Therefore, this approach can

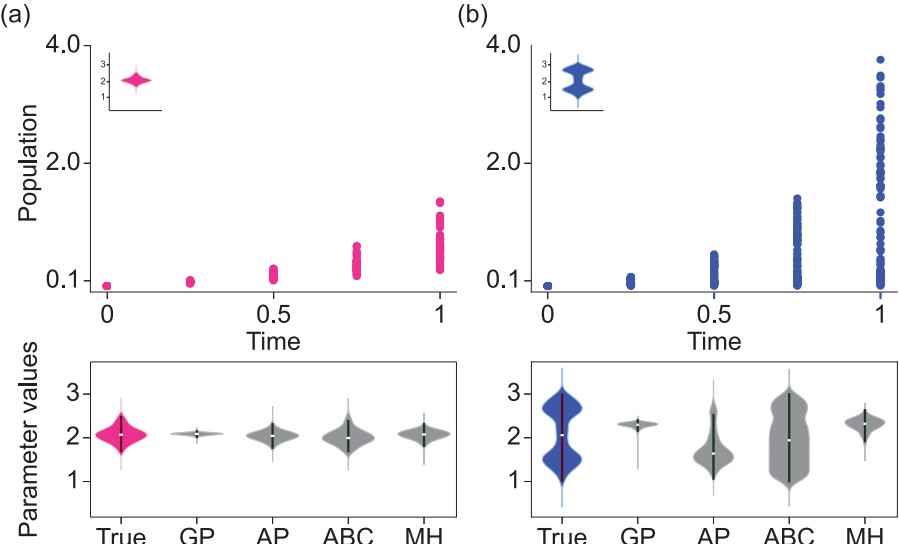

**Fig 1. Parameter estimation in the exponential growth model with Repeated Cross-Sectional (RCS) data.** An exponential growth model $y'(t) = ay(t)$ represents the amount of population, $y(t)$, changes over time, $t$. We then estimated parameter $a$ that can fit the model to a given RCS data (a-b). When the true parameter distribution of $a$ is unimodal (a, top-panel), corresponding RCS data is generated by parameters $a$, and populations per time do not diverge (a, top). In this case, previous methods, such as Gaussian Process (GP), All Possible combinations (AP), Approximate Bayesian Computation (ABC, S1 Algorithm), and Metropolis-Hastings (MH, S2 Algorithm) can estimate true parameter distributions (bottom) (a, bottom). When the true parameter distribution of $a$ is bimodal (b, top-penal), populations per time diverge (b, top). In this case, previous methods fail to estimate the shape of true parameters (b, bottom). Additionally, we performed the same estimation task with various prior distributions and observed that both ABC and MH still failed to accurately estimate the true distribution (S1 Fig).

fail when the underlying distribution is not unimodal, potentially leading to an incorrect estimation of the shape of parameter distributions and a loss of data information [18]. We also employed Bayesian-based estimation methods, such as Approximate Bayesian Computation (ABC) [19, 20] and Metropolis-Hastings (MH) [21, 22], to estimate the parameter distribution for the RCS data. However, the resulting parameter estimates can vary depending on the shape of the prior distribution [23]. Specifically, these methods fail to accurately estimate the parameter distribution, especially when applied to RCS data, leading to a loss of data information (Fig 1, see also S1 Fig).

To address this issue, this study developed the Estimation of Parameter Distribution (EPD) method, inspired by ABC, which can accurately estimate the shape of parameter distributions from RCS data in systems modeling. The proposed method can accurately and precisely determine the parameter distributions in various systems through the following three steps: First, we randomly choose one observed value for every time point, creating an artificial time trajectory. Next, we estimate the parameters **p** of the differential equation that minimizes the difference between its solution and the time trajectory, denoted by $L(\mathbf{p})$. Through this, we automatically generate the prior distribution from the RCS data and equations. Finally, by repeating the first step $N$ times, we obtain a collection of parameter sets $\mathbf{p}_n$ along with their respective differences $L(\mathbf{p}_n)$, for $n = 1, \ldots, N$. We then define the probability that each $\mathbf{p}_n$ came from the true parameter distribution based on the $\{L(\mathbf{p}_n)\}$, and draw the distribution by collecting only $\mathbf{p}_n$ selected based on their probability values. Since the selection process is based on the $\{L(\mathbf{p}_n)\}$, an additional threshold for selecting $\mathbf{p}_n$ is not required. Through this process, we show that EPD accurately captures true parameter distributions for the following models:

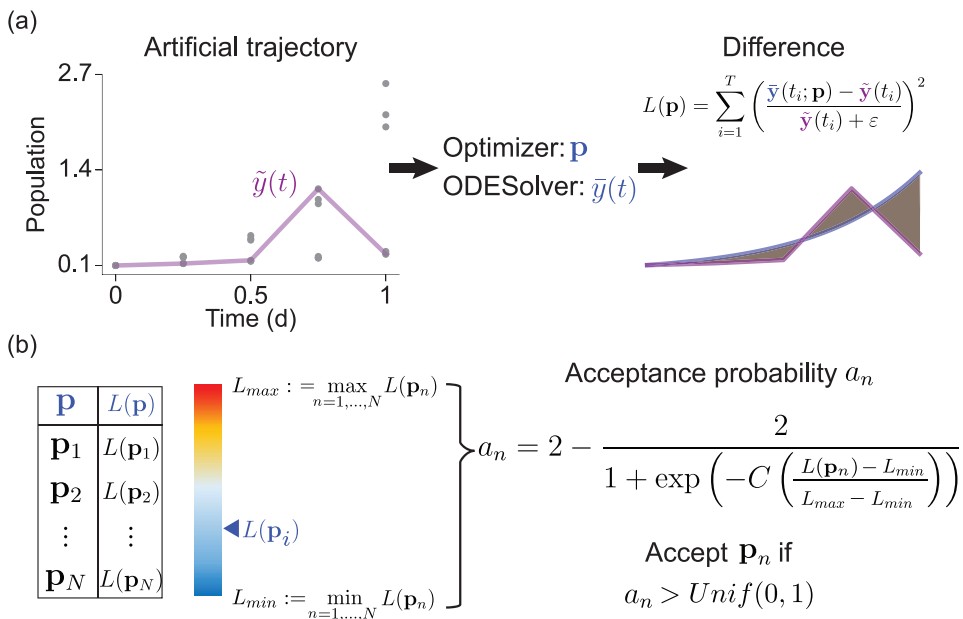

**Fig 2. Development of Estimation of Parameter Distribution (EPD) that can estimate the underlying parameter distributions for arbitrary shapes, even within the RCS data.** (a) A random sample is chosen at each timestamp and arranged in a time-ordered sequence to form a time series, $\tilde{y}(t)$. An optimizer is used to estimate the parameters of this series. Afterward, an ODE solver calculates the solution for these estimated parameters, $\bar{y}(t)$. We then define the objective function, $L(\mathbf{p})$, which measures the discrepancy between $\tilde{y}(t)$ and $\bar{y}(t)$ (right). (b) This process is repeated to generate a list of parameters and their respective errors (left). We then determine the minimum and maximum error values to calculate the probability of accepting the $n$-th parameter, $\mathbf{p}_n$, based on its error. A random number from 0 to 1 is generated uniformly. If this number is less than the calculated probability, $a_n$, then $\mathbf{p}_n$ is accepted (right).

1) exponential growth, 2) logistic population models [24], 3) target cell-limited model with delayed virus production [15, 25, 26], 4) Michaelis-Menten (Chapter 7 in [27]), and 5) Lorenz system [28].

## Results on simulation datasets

Fig 2 provides an overview of our method(EPD) for estimating parameter distributions. The EPD initially generates several artificial time series datasets based on the RCS data (Fig 2(a), left). We then evaluate the suitability of these artificial datasets by fitting them to the mathematical model (Fig 2(a), right). Subsequently, we select the artificial datasets that best match the RCS data and estimate the parameter distribution by obtaining the parameters from these optimal artificial datasets (Fig 2(b)). To verify our method, we initially used a simulation dataset. Using a numerical method, we obtained solutions corresponding to different sets of parameters. We then assumed that each solution cannot be continuously observable over time and added a Gaussian noise to account for observational errors in real-world data (i.e., RCS data). Utilizing this RCS data, the EPD captured the shape of parameter distribution.

### EPD can infer the various shapes of underlying parameter distribution of the simple exponential growth model

The exponential growth model can be used to analyze the growth patterns in population dynamics,

$$y' = ay,$$

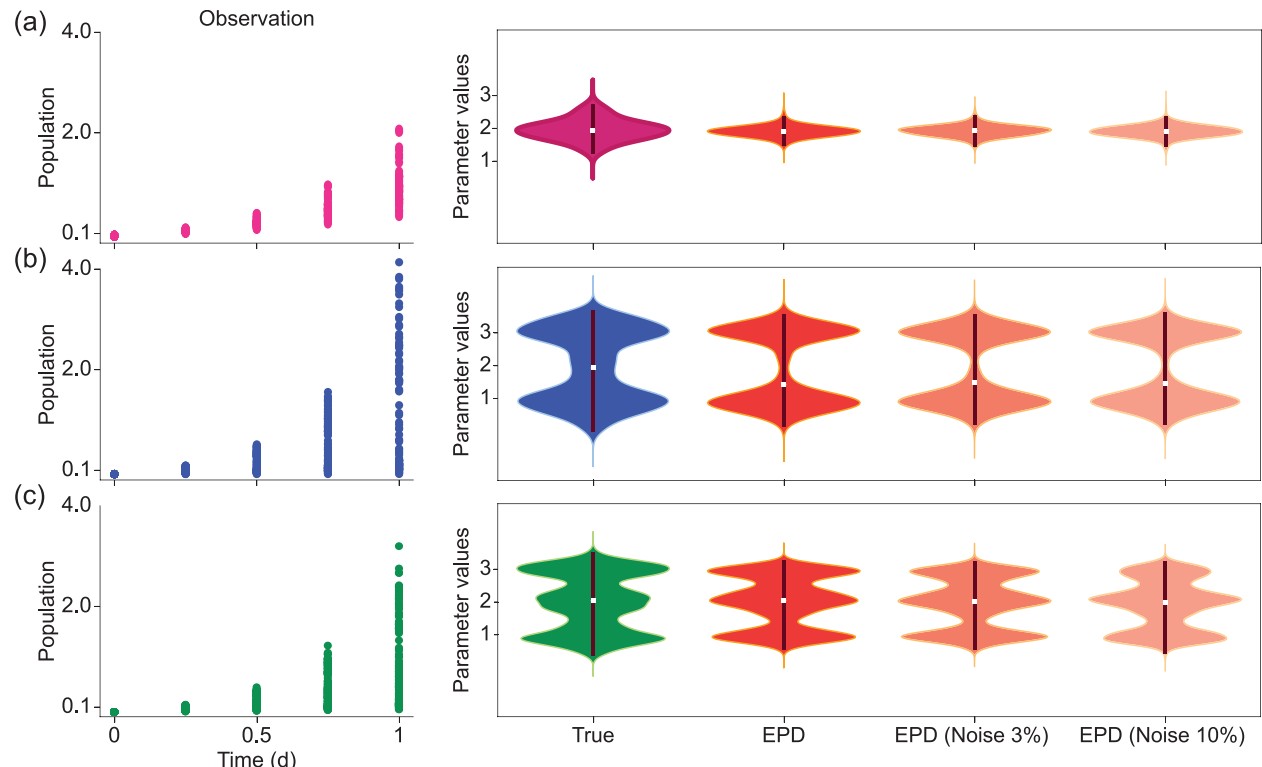

**Fig 3. Estimation of true distributions by EPD in datasets exhibiting unimodal, bimodal, and trimodal parameter distributions within the exponential growth model.** (a) When the true parameter distribution is unimodal, we applied EPD on the observed data (left) and estimated the parameters (right). Notably, EPD accurately estimated parameter distributions even when we added 3% or 10% multiplicative noise to the data (b, c). Likewise, EPD was confirmed to estimate bimodal and trimodal parameter distributions effectively.

where $y(t)$ represents the number of populations at time $t$ and $a$ is the population growth rate. We evaluated the performance of EPD in estimating true parameter distributions that reflect a given dataset through the exponential growth model. For this evaluation, we first generated a simulation dataset through a numerical solver with different five growth rates $a$ obtained from an unimodal distribution (Fig 3(a)). Specifically, the simulation dataset comprised five snapshot data at time points $t = 0, 0.25, 0.5, 0.75,$ and 1. Notably, we assume each observed value is not time-traceable, (i.e., RCS data). To apply this dataset to EPD, we generated 1,000 trajectories with observation values randomly selected at each time point. For each trajectory, we assigned an acceptance probability that reflects the likelihood of the trajectory's parameters being derived from the true parameter distribution (See Method for details). Therefore, we showed that EPD can accurately estimate the shape of true parameter distribution (i.e., unimodal distribution) (Fig 3(a), right-EPD). Furthermore, EPD also can estimate the same distribution even when the data was subjected to multiplicative noise at 3% and 10%, respectively. Subsequently, we extended the evaluation task with different datasets, reflecting different shapes of parameter distributions: a bimodal and a trimodal distribution (Fig 3(b) and 3(c), left), respectively. In each case, EPD consistently inferred the true parameter distributions even when having the noise (Fig 3(b) and 3(c), right). Therefore, these simulation results underscore that EPD accurately estimates the true parameter distributions that reflect the dataset.

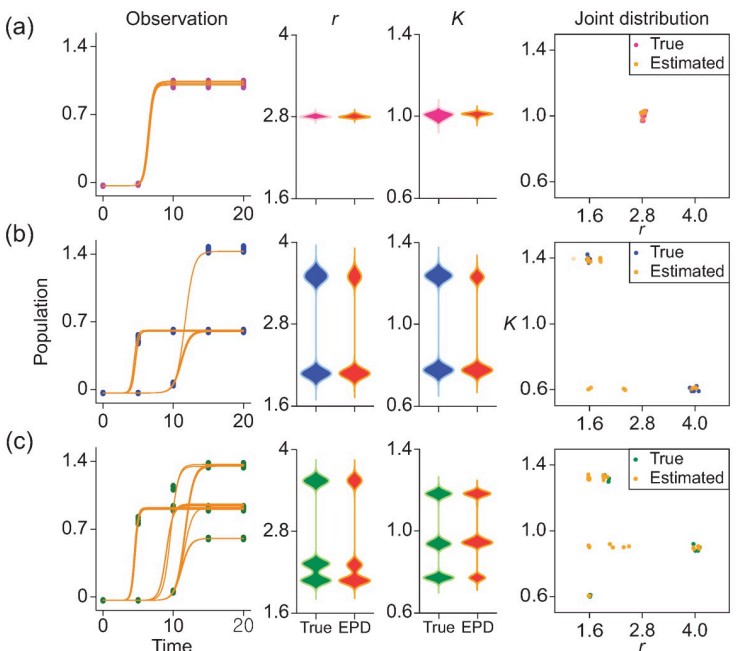

**Fig 4. Estimation of parameter distribution for the logistic growth model.** The model includes two parameters, *r* and *K*. For scenarios involving unimodal, bimodal, and trimodal distributions, we used the RCS data (left, Observation-dots) and compared true and estimated the parameter distributions (right, *r*, *K*-True, EPD) (a-c). EPD accurately estimates the true parameter distribution when the true distribution is unimodal (a). In the bimodal case, EPD also accurately captures the marginal distributions of *r* and *K*. The true parameter distribution in this case consisted of one group with a low growth rate and high capacity, and another group with a high growth rate and low capacity (b-Joint distribution, True). Interestingly, EPD not only identified these two true groups but also detected a new group with a low growth rate and low capacity (b-Joint distribution, Estimated), revealing that this problem is not fully identifiable (b). For the trimodal case, EPD accurately estimates the distributions (c). Similar to (b), we found the other group near *r* = 2.2 and *K* = 0.9 (c-Joint distribution, Estimated). To verify that the estimated parameters are reasonable, we plotted all trajectories corresponding to the estimated parameters (Observation-yellow lines). This indicates that our estimation captured all parameters that pass through the RCS data. In other words, although they may not be included in the actual joint distribution, this indicates that we have successfully recovered all potential characteristics of the data.

## Logistic population model

The logistic population model has been utilized to understand the growth dynamics of the level of protein over time *t*, *y*(*t*):

$$y' = ry(1 - y/K),$$

where *y* quantifies protein levels over time, *r* is the growth rate, and *K* represents the maximum sustainable population size that the environment can support. To evaluate the estimation performance of EPD with this model, we initially assume an unimodal distribution for the parameters, centered around the peaks of (2.8, 1.0) as a true parameter distribution. Subsequently, we generated the 12 numerical solutions for the parameter (*r*, *K*) sampled from this distribution. These 12 solutions were used to record observation data at *t* = 5, 10, 15, and 20 months, with an initial protein level of *y*(0) = 0.0001 (Fig 4(a), left). To apply this dataset to EPD, we generated 1,000 trajectories with observation values randomly selected at each time point. Similarly to the results for the first exponential model, EPD demonstrated its efficacy in accurately estimating true parameter distributions (Fig 4(a), right). For the case when the true distribution is bimodal or trimodal, we included sets of parameters near centers (4.0, 0.6), (1.6, 1.4)

and (1.6, 0.6), (4.0, 0.9), (2.0, 1.3), respectively (Fig 4(b) and 4(c), left). After we applied EPD to these datasets separately, we validated EPD can estimate the true parameter distributions regardless of the shape (Fig 4(b) and 4(c), right). Notably, EPD estimated the interpolation of two centers of true parameters, indicating that EPD can detect all possible combinations of scenarios for ($r$, $K$). That is, in terms of marginal distribution for each parameter, EPD still exhibited high levels of accuracy in predicting these distributions.

## Target cell-limited model with delayed virus production

We additionally performed a benchmark study in estimating the parameter distributions of a target cell-limited model with delayed virus production, characterized by four principal populations: susceptible epithelial cells $T$, eclipse phase $I_1$, active virus production $I_2$, and the virus population $V$.

With the four variables, the target cell-limited model can be described by the following differential equations:

$$
\begin{aligned}
\frac{dT}{dt} &= -\beta TV, \\
\frac{dI_1}{dt} &= \beta TV - \kappa I_1, \\
\frac{dI_2}{dt} &= \kappa I_1 - \frac{\delta I_2}{K_\delta + I_2}, \\
\frac{dV}{dt} &= pI_2 - cV.
\end{aligned}
\tag{1}
$$

Specifically, susceptible cells $T$ are infected by the virus proportional to $V$ with proportional constant $\beta$. Subsequently, the infected cells enter the eclipse phase $I_1$ before progressing to active virus production $I_2$ at rate $\kappa$. Virus production is regulated at a specific rate $p$ per cell, while the virus $V$ is eliminated at a clearance rate $c$, and infected cells $I_2$ are removed according to the function $I_2/(K_\delta + I_2)$, where $K_\delta$ represents the half-saturation constant and $\delta$ denote the death rate of infected cells.

To validate the predictive performance of EPD with this model, we obtained an RCS dataset for $T$, $I_1$, $I_2$, and $V$ that is generated by 12 different sets of parameters. The parameters were chosen near the center ($2.4 \times 10^{-4}$, 1.6, 13.0, 4.0, $1.6 \times 10^6$, $4.5 \times 10^5$) from [25] (Fig 5(a), left). Using 12 sets of parameters sampled from this distribution, we obtained simulation data over 12 days with initial conditions $[T(0), I_1(0), I_2(0), V(0)] = [10^7, 75, 0, 0]$. Then, we applied EPD to this dataset, predicting original parameter distributions (Fig 5(a), right). Surprisingly, even when the shape of true parameter distributions is bi- or trimodal (See Table 1 for center values), EPD can accurately estimate true parameter distributions $p$ and $K_\delta$ (Fig 5(b) and 5(c)). The estimation result of remaining parameters, $\beta$, $c$, $\kappa$, and $\delta$, were provided in (S2 Fig).

## Michaelis-Menten equation

The Michaelis-Menten model describes the time evolution of substrate concentration, $[S]$, and product concentration, $[P]$, in an enzyme-catalyzed reaction (Chapter 7 in [27]). Under the assumption of quasi-steady-state conditions, where the concentrations of the enzyme-

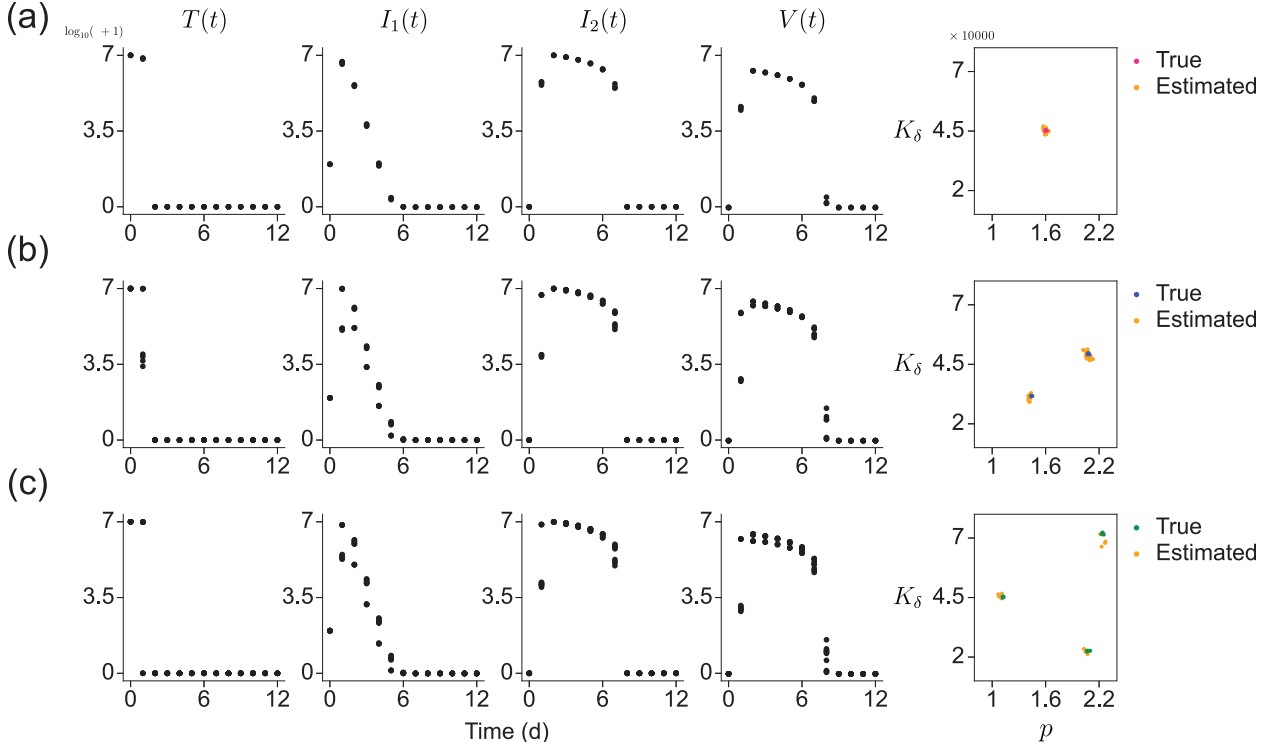

**Fig 5. Estimation results on the parameter distribution for a target cell-limited model with delayed virus production.** This model describes four components $T$, $I_1$, $I_2$, and $V$ over time. We explore three different parameter distributions: unimodal (a), bimodal (b), and trimodal (c). For each case, we used RCS data for four populations (left) and presented estimation results for two parameters $p$ and $K_\delta$ (right) among the six parameters in the model.

substrate complex and free enzyme remain constant, the system can be expressed as follows:

$$\frac{d[S]}{dt} = -\frac{K_{\text{cat}}[E_0][S]}{K_m + [S]},$$

$$\frac{d[P]}{dt} = \frac{K_{\text{cat}}[E_0][S]}{K_m + [S]},$$

where $K_{\text{cat}}$ is the catalytic rate constant, $[E_0]$ is the total enzyme concentration, and $K_m$ is the Michaelis constant. To validate the performance of EPD in the Michaelis-Menten equation, we designed three different observations (Fig 6, Observation) with a known $[E_0]$ value of $5.0 \times 10^{-4}$. Specifically, we obtained RCS data from 12 trajectories of $S(t)$ and $P(t)$ which were

**Table 1. Parameter values for different distribution types within the target cell-limited model using synthetic data.**

| Distribution type | $\beta$ (×10$^{-4}$) | $p$ | $c$ (×10$^1$) | $\kappa$ | $\delta$ (×10$^6$) | $K_\delta$ (×10$^4$) |
|---|---|---|---|---|---|---|
| Unimodal | 2.40 | 1.60 | 1.30 | 4.00 | 1.60 | 4.50 |
| Bimodal | 2.88 | 1.44 | 1.82 | 5.20 | 1.28 | 3.15 |
| | 2.16 | 2.08 | 0.91 | 3.20 | 1.76 | 4.95 |
| Trimodal | 2.88 | 1.12 | 1.56 | 4.00 | 1.44 | 4.50 |
| | 1.68 | 2.24 | 1.82 | 5.60 | 1.60 | 7.20 |
| | 2.16 | 2.08 | 0.78 | 2.40 | 1.92 | 2.25 |

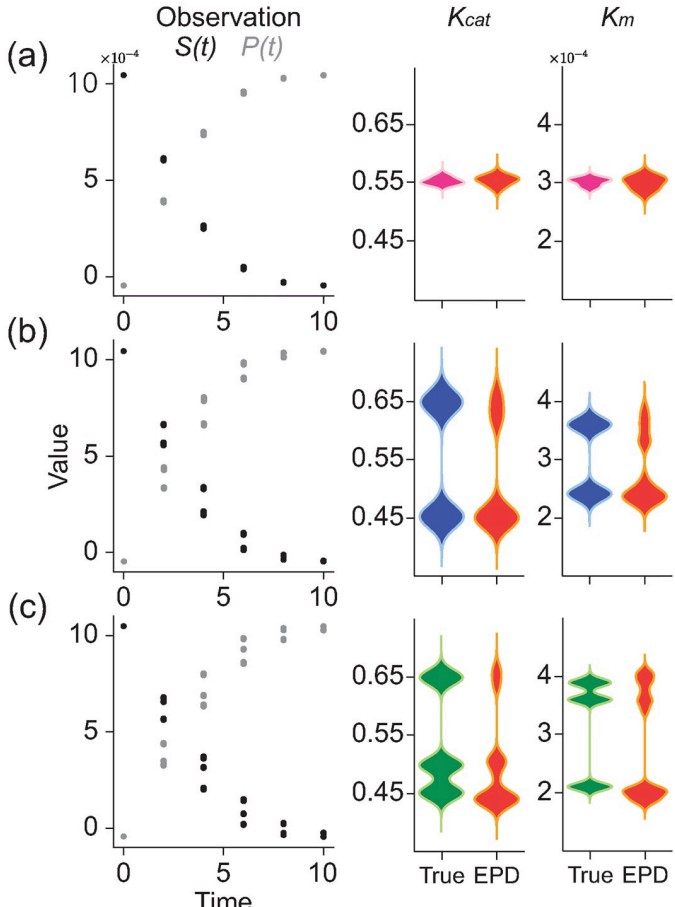

**Fig 6. Estimation of parameter distribution for the Michaelis-Menten equation.** Using a numerical solver, we generated RCS data from 1000 solutions of the Michaelis-Menten equation (left, $S(t)$, and $P(t)$). We then estimated the distributions of $K_{cat}$ and $K_m$ through EPD (right). For the unimodal (a), bimodal (b), and trimodal (c) scenarios, we found that EPD accurately captured the underlying parameter distributions.

generated through the true parameter distribution with a single peak value at $(K_m, K_{cat}) = (3.0 \times 10^{-4}, 0.55)$ (Fig 6(a), Observation). This pair is typically observed for pepsin [29]. Similarly, once peak values were selected as $(K_m, K_{cat}) = (2.7 \times 10^{-4}, 0.44)$ and $(3.9 \times 10^{-4}, 0.66)$ for the bimodal distribution, and $(K_m, K_{cat}) = (2.7 \times 10^{-4}, 0.39)$, $(3.9 \times 10^{-4}, 0.66)$, and $(3.0 \times 10^{-4}, 0.72)$ for the trimodal distribution, respectively, we generated 12 trajectories for each peak, which were also considered as RCS data (Fig 6(b) and 6(c), Observation). We then estimate the parameter distributions using EPD, indicating that EPD accurately estimates true distributions (Fig 6(a), 6(b) and 6(c), $K_{cat}$, $K_m$-EPD).

## Lorenz system

The following Lorenz system describes a simple atmospheric circulation using three key variables [28]: rate of convective motion in the system $X$, temperature difference between the ascending and descending flows within the convection cell $Y$, and deviation of the system from

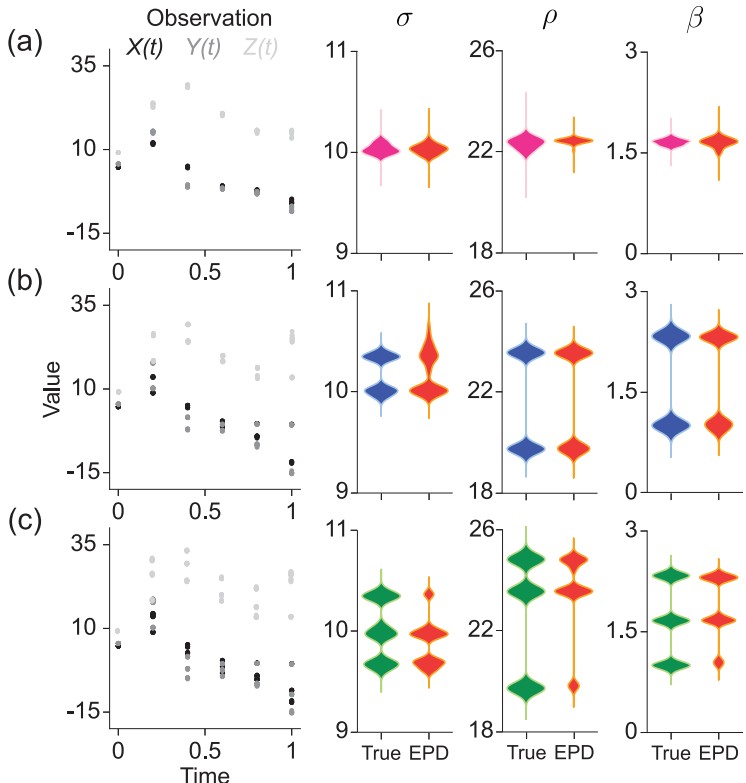

**Fig 7. Estimation of parameter distribution for the Lorenz system.** Using a numerical solver, we generated RCS data from 1000 solutions of the Lorenz equation (left, $X(t)$, $Y(t)$, and $Z(t)$). We then estimated the distributions of $\sigma$, $\rho$, and $\beta$ through EPD (right). For the unimodal (a), bimodal (b), and trimodal (c) scenarios, we found that EPD accurately captured the underlying parameter distributions.

thermal equilibrium or the vertical temperature distribution in the convection $Z$:

$$\frac{dX}{dt} = \sigma(Y - X),$$

$$\frac{dY}{dt} = X(\rho - Z) - Y,$$

$$\frac{dZ}{dt} = XY - \beta Z,$$

where Prandtl number, $\sigma$, controls the ratio of fluid viscosity to thermal diffusivity, Rayleigh number, $\rho$, drives convection based on temperature differences, and $\beta$ is related to the damping of convection. To validate the predictive performance of EPD with this system, we obtained a RCS dataset for $X$, $Y$, and $Z$ that is generated by 12 different sets of parameters (Fig 7, Observation). In particular, in the unimodal case, the true parameter values ($\sigma$, $\rho$, $\beta$) were selected near the peak values (10, 22.5, 1.67). For the bimodal case, two different pair of peak values were chosen as ($\sigma$, $\rho$, $\beta$) = (10.5, 18.0, 1.0) and (9.5, 27.0, 1.67). In the trimodal case, the three peak values included those used in the bimodal case, with an additional peak at (10.0, 24.75, 1.4). For 12 sets of parameters chosen from these peak values, corresponding trajectories were derived using a numerical solver (LSODA algorithm), and this data was treated as RCS data. By observing the values of ($X$, $Y$, $Z$) at six distinct time points, EPD generated 1000 artificial trajectories to estimate the true parameter distribution. Similar to previous results, EPD

accurately captured the shape of true parameter distributions when they had unimodal, bimodal, or trimodal forms Fig 7.

## Results on real-world datasets

### Logistic population model

We fitted the logistic model to amyloid-$\beta$ 40 (A$\beta$40) and amyloid-$\beta$ 42 (A$\beta$42) datasets, utilizing them as biomarkers for diagnosing dementia [24, 30, 31]. In the experimental datasets, the number of (A$\beta$40) and (A$\beta$42) were recorded at four different time points at 4, 8, 12, and 18 months, and each time point had 12–13 independent observation samples (Fig 8(a) and 8(b), left). We then normalized the levels of (A$\beta$40) and (A$\beta$42) (measured in picograms per milliliter), so that the peak value observed in 12-month-old mice was set to 1.0. As the data is RCS type, we utilized EPD for inferring the shape of parameter distribution. Our results indicated significant heterogeneity in the growth dynamics of amyloid beta, as demonstrated by distinct centers of parameters for both growth rates and population capacities (Fig 8(a) and 8(b), right). As the heterogeneity that shows the progression of amyloid beta accumulation can vary significantly across different population subsets, the estimated parameter distribution demonstrates the importance of personalized diagnostic and therapeutic strategies in combating dementia. Furthermore, we observed that no single parameter could effectively account for the trajectories that included data points at the high rate of amyloid A$\beta$42 at 8 months or A$\beta$40 at

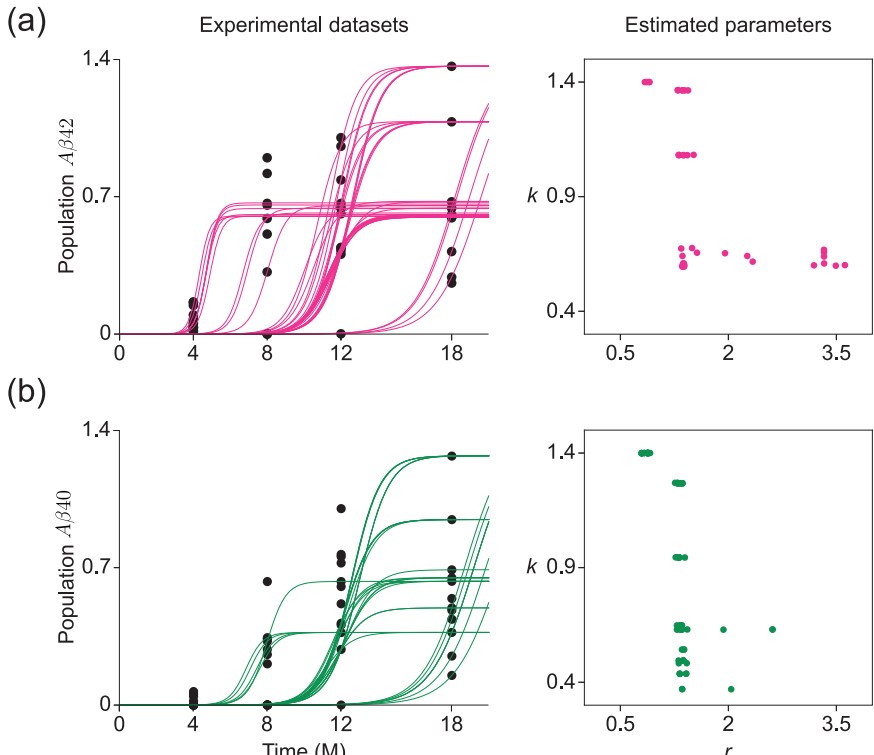

**Fig 8. Parameter estimation results of logistic model with real experimental datasets on amyloid beta accumulation.** (a-b) The accumulation of amyloid beta 40 and 42 were estimated at 4, 8, 12, and 18 months (left black dots). With these datasets, we calculated solutions of logistic model (red and green curves on the left), corresponding to estimated parameters obtained through EPD (right).

12 months, thereby indicating that when the mouse reaches a certain rate of amyloid A$\beta$40 or 42 early in their life, it cannot survive for a long time.

## Target cell-limited model

We fitted the target cell-limited model to the virus dataset, obtained from [25]. In the dataset, daily viral loads $V$ were measured from groups of BALB/cJ mice infected with influenza A/ Puerto Rico/8/34 (H1N1) virus (PR8) (Fig 9, left). The mice received an intranasal administration of a dose of 75 TCID50 of the PR8 virus at the initial time point (t = 0), where TCID50 is the concentration required to infect 50% of the cell cultures [32, 33]. Unlike the previous estimation tasks, only the value of V is observable out of all the populations in the model, thus the parameter estimation was performed using only the viral loads. Data was collected over 12 days, with 10 animals sampled per time point. For faster computation, we utilized four time points observed at 1, 3, 7, and 8 days. With this RCS data, we applied EPD for estimating the parameter distributions with a target cell-limited model with delayed virus production (Fig 9, right). Surprisingly, this result showed that two distributions of the parameters $p$ and $K_\delta$ have at least three centers. The estimation result of the remaining parameters, $\beta$, $\kappa$, $c$, and $\delta$, were provided in (S3 Fig). Therefore, EPD not only can accurately predict parameters, but also demonstrate the heterogeneity of parameters. Thus, when it is near the value from the previous research, our results suggest the existence of multiple parameter sets that can represent this dataset, beyond those previously identified parameters.

## Discussion

This study proposed the EPD method for inferring parameter distributions from Repeated Cross-Sectional data in systems modeling. Unlike previous approaches, which often overlooked data heterogeneity and resulted in information loss, EPD determines more precise and accurate parameter distributions across a variety of systems. By estimating parameter distributions, EPD facilitates a deeper understanding of the underlying dynamics of these systems.

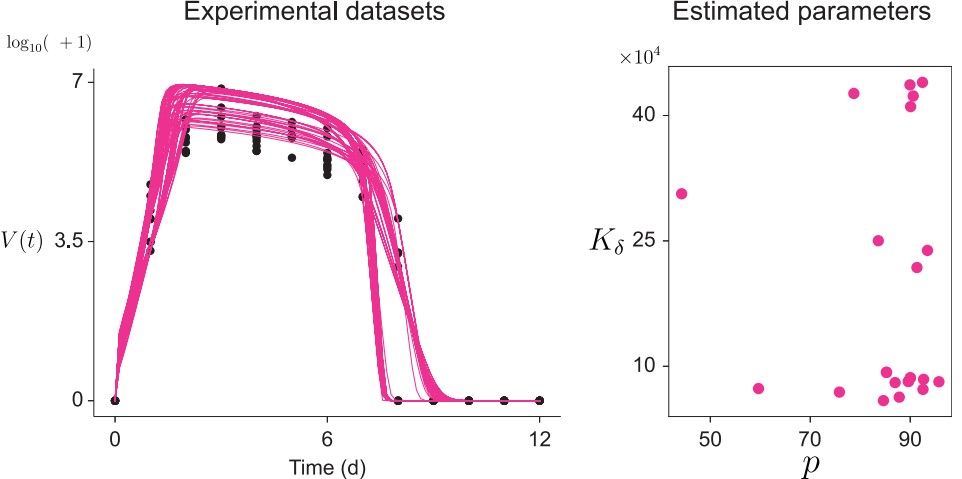

**Fig 9. Parameter estimation results of a target cell-limited model with delayed production with a real experimental virus infection dataset.** Black dots represent the virus population $V$ over time $t$, the only measurable factor among the four components in the model (left). Artificial trajectories using EPD were simultaneously provided on the same graph (left, red curves). Two parameters corresponding to the artificial trajectories, $p$ and $K_\delta$, were estimated (right, red dots). The other parameters, $\beta$, $c$, $\kappa$, and $\delta$, are detailed in S3 Fig.

Consequently, this paper not only advances our capacity to model and predict system behaviors more effectively, but also highlights the critical need to account for data variability and distribution when analyzing complex systems.

To develop EPD, we utilized the relative error when constructing the objective function $L$($\mathbf{p}$), Eq 4, which measures the difference between the artificial trajectories and the corresponding solutions of Eq 2. By maintaining a similar scale across all samples over time, this approach allows for unbiased estimation, independent of the absolute magnitude of the data. To demonstrate this, we conducted a parameter estimation using EPD with the absolute error in $L(\mathbf{p})$ (i.e., $L(\mathbf{p}) = \sum_{i=1}^{T} |\bar{\mathbf{y}}(t_i; \mathbf{p}) - \tilde{\mathbf{y}}(t_i)|^2$). Unlike the relative error, the absolute error can yield two crucial issues in the optimization step for finding $\mathbf{p}$. First, the varying scale of magnitude in observation points over time $t$ introduces errors in the target cell limited model (S4 Fig). In this example, progression rate $\kappa$ and death rate $\delta$ are mainly determined by the period when the concentration of infected sell $I_2$ reached the highest value (i.e., $I_2 \sim 10^7$, S4(a) Fig-$I_2$). Conversely, the parameters $p$ and $c$ are mainly affected by the balance between virus production and clearance, especially during the low viral load stages. (i.e., $I_2 = 0$ and $V \sim 0$, S4(a) Fig-$I_2$, $V$). Therefore, optimization steps for determining the minimizer of the $L(\mathbf{p})$ are mainly conducted by finding the suitable $\kappa$ and $\delta$ as the corresponding absolute error terms in $L(\mathbf{p})$ are dominant. Second, we found the bias in the values of $L(\mathbf{p})$ in the EPD algorithm, stemming from the absolute magnitude of RCS data at each fixed time point, leading to biased parameter estimates. To verify this, we calculate $L(\mathbf{p}_i)$ using Fig 3(b)-Observation, for $i = 1, \ldots, N$ with the sum of squares, then divided the distribution of $L(\mathbf{p}_i)$ values into two groups: one group (blue) selecting the smallest observation at the final time point ($t = 1$) and the other group (red) selecting the largest observation (S5 Fig). Consequently, the parameter estimates from the red graph, which have relatively larger values of $L(\mathbf{p})$, exhibit lower acceptance probabilities, leading to a reduction in the magnitude of the corresponding peak in the estimated parameter distribution.

We also utilized the logistic transformation in the acceptance probability. While existing methods (such as ABC) use a threshold for the final decision, this can result in the shape of estimated parameter distribution varying depending on the threshold value (S6 Fig). To address this issue, we used the logistic transform, allowing the final decision to be made based on a random number between 0 and 1 instead of relying on a threshold. Following this transformation, the resulting value can consistently be used as the acceptance probability for any given dataset.

Determining the appropriate scaling factor $C > 0$ in the acceptance probability (Fig 2(b)) is important for EPD, as it influences the likelihood of accepting a given parameter $\mathbf{p}$. For example, a higher positive scaling factor leads to only accepting parameters that result in lower objective function values. While a large scaling factor might suggest that EPD estimations become more precise by focusing on lower loss values, we must be cautious with its magnitude, especially when dealing with parameter heterogeneity (Fig 10). Although careful selection of the scaling factor is required, EPD can accurately estimate the underlying distributions across various differential equations using scaling factor values between 100 to 300 (Table 2).

Despite these implementations, we observed a bias due to the missing observation data. To investigate this further, we conducted additional experiments to examine how the shapes of the parameter distributions obtained from EPD are affected by the missing data in the exponential model (Fig 3(a), 3(b) and 3(c), left). In particular, we removed 90% of the observation data at specific time points (S7 Fig, left—Dropped points). As the amount of missing data increased, the estimation results using EPD exhibited a bias, leading to variations in peak heights (S7 Fig, right). Notably, although the missing data introduces bias in the peak heights,

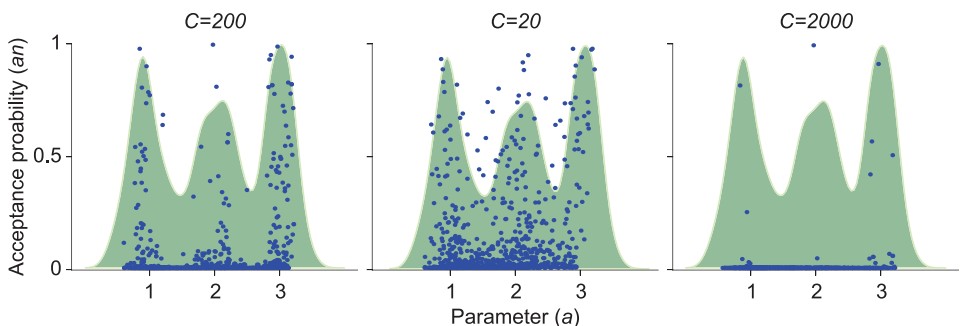

**Fig 10. Distribution of the acceptance probability $a_n$ with different scaling factors $C$ in the exponential model with a trimodal parameter distribution (green).** We first calculate the parameter $\mathbf{p}_n$ corresponding to n-th artificial trajectory $\bar{\mathbf{y}}_n$ (Fig 2(b), left), then plotted $(\mathbf{p}_n, a_n)$ (blue dots), for $n = 1, \ldots, N$. When $C = 200$, only high values $a_n$ are located near the peaks of the true parameter distribution (green), leading to correct parameter estimates (left). However, when $C$ is too small ($C = 20$), high values of $a_n$ are widely spread across the entire parameter range (middle). Conversely, when $C$ is too large ($C = 2000$), the overall acceptance probabilities are concentrated near 0, resulting in poor parameter estimates (right).

EPD still estimates the overall shape of the parameter distribution, reflecting the heterogeneity of the RCS data.

We have several limitations for future directions. Because EPD utilized ODESolver, we need to choose a suitable ODESolver that can solve the given dynamical system (differential equation). Second, computational cost arises when calculating the solution through a numerical solver and when fitting the solution through optimization techniques. As parameters were selected from a large set of synthetic trajectories that were obtained from the numerical solver, the computational costs became substantial, since it is proportional to both the complexity of equations and the number of trajectories (See Table 3). For future research, we are considering the use of a machine learning-based model emulator (such as Physics-informed Neural Networks incorporating hypernetworks [34] or DeepONet [35–38]]) to handle larger systems with RCS data more efficiently. Specifically, these emulators, which are pre-trained to approximate system solutions and parameters simultaneously, we can expect to significantly reduce the computational costs associated with these tasks. Last, we found the identifiable problem through the logistic example (Fig 4). In this example, data is based on assumptions corresponding to two biological scenarios: one with a low growth rate and high capacity and the other with a high growth rate and low capacity. In the estimation results obtained through EPD, we identified not only the two true cases but also a new group with low growth rate and capacity (Fig 4, Joint distribution). This is natural because the RCS data lost all continuous information regarding growth and capacity, leading to possible outcomes. Although we have presented only the raw estimation results to understand the RCS data, this issue can be

**Table 2. Scaling factors for the EPD corresponding to each equation and dataset.** The scaling factor $C$ was selected within the range of 100 to 300 across the five different types of equation, to estimate distribution that closely matches the true distribution. For the real-world dataset, the scaling factor was determined based on the most prevalent value observed in the simulation datasets.

| Type | Exponential | Logistic | Target Cell Limited Model | Lorentz | Michaelis Menten |
|---|---|---|---|---|---|
| Unimodal | 200 | 100 | 300 | 100 | 300 |
| Bimodal | 200 | 100 | 300 | 100 | 300 |
| Trimodal | 200 | 200 | 300 | 300 | 300 |
| Real | - | 200 | 300 | - | - |

**Table 3. Computational Cost for the EPD corresponding to each equation and dataset.** The running time is measured based on several factors, including the number of equations ($n_y$), number of parameters ($n_p$), number of time points ($T$), number of data points for each time point, ($J_i$), total number of data points ($N$), and number of artificial trajectories ($N$). The results indicate that the computational cost is directly proportional to $n_y$, $n_p$, $T$, and $N$, while showing a small correlation with the underlying parameter distribution.

| Equation | Parameter Distribution | $n_y$ | $n_p$ | $T$ | $J_i$ | $N$ | Running Time (sec) |
|---|---|---|---|---|---|---|---|
| Exponential model | Unimodal | 1 | 1 | 5 | 50 | 1000 | 4.06 |
| | Bimodal | | | | 100 | | 4.18 |
| | Trimodal | | | | 150 | | 4.26 |
| Exponential model w noise | Unimodal | 1 | 1 | 5 | 50 | 10000 | 80.75 |
| | Bimodal | | | | 100 | | 83.62 |
| | Trimodal | | | | 150 | | 84.18 |
| Logistic model | Unimodal | 1 | 2 | 5 | 12 | 3000 | 162.70 |
| | Bimodal | | | | 12 | | 181.54 |
| | Trimodal | | | | 12 | | 202.69 |
| | Real-world | | | | 12–13 | | 188.85 |
| Target Cell Limited model | Unimodal | 4 | 6 | 13 | 12 | 1000 | 4555.18 |
| | Bimodal | | | | 12 | | 9036.38 |
| | Trimodal | | | | 12 | | 12318.30 |
| | Real-World | | | | 10 | 10000 | 111667.74 |
| Lorentz System | Unimodal | 3 | 3 | 6 | 12 | 1000 | 28.90 |
| | Bimodal | | | | 24 | | 39.09 |
| | Trimodal | | | | 36 | | 42.62 |
| Michaelis-Menten | Unimodal | 2 | 2 | 6 | 12 | 1000 | 9.50 |
| | Bimodal | | | | 24 | | 11.39 |
| | Trimodal | | | | 36 | | 10.80 |

resolved in actual estimations by adding constraints that prevent biologically implausible outcomes.

Finally, we consider other transformations when constructing acceptance probability. We currently apply the logistic transformation to the loss function. Without any transformation, the acceptance probability is directly proportional to the loss function value, which has no significant distinction between parameters with different loss values. The logistic transformation helped EPD to select a parameter with much less objective function value. In future research, we will clarify whether this transformation is optimal by providing rigorous proof or conducting various experiments with other transformations.

## Materials and methods

### The parameter estimation problem

We propose the method for estimating the distribution of parameters within the time evolutionary differential equation (ODE), represented as:

$$\mathbf{y}'(t) = f[\mathbf{y}(t), \mathbf{p}, t], \tag{2}$$

where $\mathbf{y} = \mathbf{y}(t) \in \mathbb{R}^{n_y}$ represents the nonnegative population size with dimension $n_y$ at time $t$. The set of parameters $\mathbf{p} \in \mathbb{R}^{n_p}$ encompasses biological or physical properties such as the growth rate or the carrying capacity for the population, respectively. The problem is to estimate the distribution of the parameter $\mathbf{p}$ when the observation data corresponding to $\mathbf{y}$, denoted by $Y$, is provided as the RCS data (Fig 2(a), left). In particular, $Y$ includes a set of $J_i$ observed data points at each time point $t_i$, $Y_i$, for $i = 1, 2, \cdots, T$, where $T$ is the total count of

time steps, i.e.,

$$Y_i = \{\mathbf{y}_1(t_i), \mathbf{y}_2(t_i), ..., \mathbf{y}_{J_i}(t_i)\}.$$

Because the data $Y$ not only exhibits different observation values at each time $t_i$ but are also independent (i.e., RCS data); that is, each element in $Y_i$ can correspond to different parameter values $\mathbf{p}$. Therefore, our goal is to estimate the distribution of the parameter $\mathbf{p}$, rather than determining a single fixed value.

## Development of EPD, estimating the distribution of parameters

To estimate parameters corresponding to RCS data, we construct $N$ artificial trajectories, denoted as $\{\tilde{\mathbf{y}}_n\}_{n=1}^N$, aligned with specific time points $\{t_i\}_{i=1}^T$ (Fig 2(a), right). Specifically, for each $t_i$, we randomly chose one observation value, $\mathbf{y}_j(t_i) \in Y_i$, for some $1 \leq j \leq J_i$. We assume this selection probability to be uniform. We then consider $\tilde{\mathbf{y}} = \{\mathbf{y}_j(t_i)\}_{i=1}^T$ as the artificial time trajectory. By repeating this process $N$ times, we can obtain $N$ artificial time trajectories $\{\tilde{\mathbf{y}}_n\}_{n=1}^N$. Remarkably, the likelihood of choosing the trajectory $\tilde{\mathbf{y}}_n(\cdot)$, given the observations $\{Y_i\}_{i=1}^T$, can be formulated as:

$$
\begin{aligned}
P(\{\tilde{\mathbf{y}}_\mathbf{n}|\{Y_i\}_{i=1}^T\}) &= P(\tilde{\mathbf{y}}_\mathbf{n}(t_1) = \mathbf{y}_{j_1}(t_1), \tilde{\mathbf{y}}_\mathbf{n}(t_2) = \mathbf{y}_{j_2}(t_2), \ldots, \tilde{\mathbf{y}}_\mathbf{n}(t_T) = \mathbf{y}_{j_T}(t_T)) \\
&= \Pi_{i=1}^T P(\tilde{\mathbf{y}}_\mathbf{n}(t_i) = \mathbf{y}_{j_i})) = \Pi_{i=1}^T \frac{1}{J_i}.
\end{aligned}
\tag{3}
$$

where $1 \leq j_i \leq J_i$, for all $i = 1, \ldots, T$. Therefore, to sample all possible artificial trajectories, $N$ can be chosen to be proportional to the inverse of Eq 3.

Next, we estimate the set of parameters $\mathbf{p}_n$ corresponding to $n$-th artificial trajectory $\tilde{\mathbf{y}}_n$. Specifically, given $\tilde{\mathbf{y}}_n$, we first obtain the solution of Eq 2 with an initial set of parameters $\mathbf{p}$, $\bar{\mathbf{y}}_n(t; \mathbf{p})$, through LSODA algorithm [39, 40]. We next adjust the $\mathbf{p}$ so that $\bar{\mathbf{y}}_n$ can be closed to $\tilde{\mathbf{y}}_n$. This adjustment can be conducted by minimizing the following objective function, defined by a relative error between $\tilde{\mathbf{y}}_n$ and $\bar{\mathbf{y}}_n$:

$$
L_n(\mathbf{p}) = \sum_{i=1}^T \left| \frac{\bar{\mathbf{y}}_n(t_i; \mathbf{p}) - \tilde{\mathbf{y}}_n(t_i)}{\tilde{\mathbf{y}}_n(t_i) + \varepsilon(\tilde{\mathbf{y}}_n(t_i))} \right|^2,
\tag{4}
$$

where $\varepsilon = \varepsilon(\tilde{\mathbf{y}}_n(t_i)) > 0$ is a regularization constant, preventing the denominator from becoming zero in the calculation of the relative error. For the choice of $\varepsilon$, we adaptive set the values as:

$$
\varepsilon = \begin{cases} 10^{-1} & \text{if } |\tilde{\mathbf{y}}_n(t_i) = 0|, \\ 10^{-10} & \text{otherwise.} \end{cases}
$$

We then find the minimizer $\mathbf{p}_n$ of $L_n(\mathbf{p})$ using LMFIT [41] package in Python to apply the least square algorithm. The choice of using relative error is also discussed in the Discussion section.

For each $1 \leq n \leq N$, we cannot guarantee that $\tilde{\mathbf{y}}_n$ represents real continuous observation trajectories, as they were generated randomly. Hence, we cannot ensure that the minimizer $\mathbf{p}_n$ of $L_n$ with $\tilde{\mathbf{y}}_n$ in Eq 4 is an accurate estimate. Rather than treating $\{\mathbf{p}_n\}_{n=1}^N$ as final estimates, we consider them as samples from the prior distribution. That is, it is essential to assess whether the minimizer $\mathbf{p}_n$ is a reasonable parameter estimate for the RCS data. For this determination, we create the acceptance probability $a_n$, motivated by rejection sampling, which depends on how well the model (Eq 2) fits with the estimated parameters $\mathbf{p}_n$ (Fig 2(b)). Specifically, the

probability $a_n$ is calculated via a logistic transformation applied to the previously computed residuals $L(\mathbf{p}_n)$ (representing the goodness of fit) as:

$$a_n = 2 - \frac{2}{1 + \exp\left(-C\left(\frac{L(\mathbf{p}_n) - \min_n L(\mathbf{p}_n)}{\max_n L(\mathbf{p}_n) - \min_n L(\mathbf{p}_n)}\right)\right)},$$

where $L(\mathbf{p}_n)$ denotes the objective function values in Eq 4 for each fit, and $C > 0$ represents a scaling factor that can be adjusted for improved accuracy to determine how many of the parameter estimates we accept. For instance, if $C$ is close to zero, most estimates will be accepted, while if $C$ approaches infinity, most estimates will be rejected. (See also Fig 10). We have provided all the $C$ values used in this study in Table 2.

In contrast to MCMC, which iteratively refines parameter estimates to converge on the posterior distribution, our method decides on accepting or rejecting parameter sets based on their computed likelihood after explicitly minimizing a predefined objective function. Specifically, a parameter set $\mathbf{p}_n$ is accepted if it satisfies:

$$a_n > u_n \quad \text{where} \quad u_n \sim Unif(0, 1),$$

where a set $\{u_n\}_{n=1}^{N}$ is independently sampled from an identical uniform distribution over $n$. It ensures a probabilistic assessment of parameter set acceptance based on their respective goodness of fit. Note that when $C$ equals to zero, all estimated parameters will be accepted. This case will be referred to as All Possible combinations (AP) because it considers every estimated result without further refinement.

## Evaluating EPD performance in estimating parameter distributions using simulation datasets

We evaluated the performance of EPD in estimating the distribution of parameters with simulation data. We employed five distinct dynamical systems: an exponential growth model for detecting cell dynamics heterogeneity, a logistic regression model for simulating protein generation [24], a target cell-limited model for understanding virus infection dynamics [42, 43], the Michaelis-Menten equation for analyzing enzyme-catalyzed reactions [27], and the Lorenz system for studying simple atmospheric circulation [28]. With these evaluations, we show the adaptability and robust potential of EPD in accurately identifying the true parameter distributions and in forecasting system behaviors even in the presence of noise. To generate the distribution of $\mathbf{p}$ synthetically, we first consider $H$ distinct centers $\{\mathbf{p}_h^{center}\}_{h=1}^{H}$ which imply the peaks of parameters across various clusters. The large value of $H$ can pose parameter heterogeneity. We conduct uniform random sampling independently within pre-established bounds to generate parameter distribution around these centers. Specifically, we randomly select $S$ values for the parameters $\{\mathbf{p}_{(h-1)S+i}\}_{i=1}^{S}$ around $\mathbf{p}_h^{center}$ within their respective bounds as follows:

$$\mathbf{p}_{(h-1)s+i} \sim Unif((\mathbf{p}_L)_h, (\mathbf{p}_U)_h),$$

where $Unif((\mathbf{p}_L)_h, (\mathbf{p}_U)_h)$ represents the uniform distribution between $(\mathbf{p}_L)_h$ and $(\mathbf{p}_U)_h$. This results in $HS$ sampled parameter sets, which are utilized to construct trajectories and generate RCS data related to diverse biological experiment scenarios. To generate synthetic RCS data, we resolve the ODE in Eq 2 for each parameter set $\mathbf{p}$. Using the LSODA algorithm, which can adjust the balance between stiff and non-stiff structures of solutions, with initial conditions $\mathbf{y}$ (0) at time $t = 0$,

$$\tilde{\mathbf{y}}(t; \mathbf{p}) = \mathbf{y}(f, \mathbf{y}(0), t; \mathbf{p}), \mathbf{p} \in \{\mathbf{p}_1, \dots, \mathbf{p}_{HS}\}.$$

The initial value $\mathbf{y}(0)$ is determined by the experimental setup or an existing dataset. The above ODE solving mechanism yields totally $HS$ trajectories which will be designated as RCS data. Specifically, it is assumed that we can only access $HS$ data points at each observational time point $t_i$ for $i = 1, 2, \ldots, T$, where $T$ is the total number of time points, instead of a set of fully connected trajectories.

## Supporting information

**S1 Fig. Parameter estimation results for the Exponential model using Approximate Bayesian Computation (ABC) and Metropolis-Hastings (MH), based on different shapes of the prior distribution (uniform, unimodal, bimodal, and trimodal).** We estimate three True parameter distributions (unimodal (True-red), bimodal (True-blue), and trimodal (True-green)) corresponding to each dataset in Fig 1. Without any assumptions about the prior distribution, EPD can accurately estimate the True parameter distributions (EPD-orange). (a-c). When the True parameter distribution is unimodal, all estimation results closely approximate the true distribution, except for ABC with a bimodal prior distribution (a). However, when the true distribution is bimodal, all estimation results using ABC and MH show different patterns from the true distribution, except for ABC with a bimodal prior distribution (b). Similar to (b), when the true distribution is trimodal, all estimation results using ABC and MH show different patterns from the true distribution, except for ABC with a trimodal prior distribution (c). These results indicate that the ABC method estimates the similar shape of parameter distribution to the prior distribution, leading to inaccurate parameter estimates with inaccurate choice of prior distributions. (PNG)

**S2 Fig. Estimation results for the parameter distribution within a target cell-limited model based on synthetic data.** We estimated the distributions of $\beta$, $c$, $\kappa$, and $\delta$, corresponding to data in (Fig 5, left $T(t) - V(t)$), respectively. First, we applied EPD to data generated from parameters that share similar scales (Fig 5(a), left $T(t) - V(t)$). Consequently, EPD can accurately estimate the parameters (Left). Furthermore, with data generated from parameters with different scales (Fig 5(b) and 5(c), left), EPD can infer the true parameter distributions (Middle and Right, respectively). That is, EPD can estimate the true distribution of parameters even when they do not follow the normal distribution. Notably, the prediction does not contain the interpolation of the centers as not previously in the logistic model. (PNG)

**S3 Fig. Estimation results for a real experimental virus infection dataset.** We estimated the four parameters $\beta$, $c$, $\kappa$, $\delta$ that fit the target cell-limited model to the given RCS data [25]. We discovered that the estimated parameter distributions contain heterogeneity for all parameters, similar to the estimates for $p$ and $K_\delta$. Unlike previous results in [15], our findings do not follow the normal distribution shape. Nevertheless, these predictions could be a reasonable guess because they can reconstruct the trajectories through Eq 1. (PNG)

**S4 Fig. Comparison of estimation results using EPD with two different objective functions: Absolute error and relative error (Ours) in a target cell-limited model.** (a) We generated trajectories for $T(t)$, $I_1(t)$, $I_2(t)$, and $V(t)$ corresponding to one set of six parameters $(p, K_\delta, \beta, c, \kappa, \delta) = (2.4 \times 10^{-4}, 1.6, 13.0, 4.0, 1.6 \times 10^{6}, 4.5 \times 105)$ within the target cell-limited model. (b) We compared the parameter estimates obtained using the absolute error (i.e., $L_n(\mathbf{p}) = \sum_{i=1}^{T} \left| \bar{\mathbf{y}}_n(t_i; \mathbf{p}) - \tilde{\mathbf{y}}_n(t_i) \right|^2$, to those obtained using the relative error $L_n(\mathbf{p})$ in (Eq 4) (Blue and red box plots, respectively). The box plot results reveal that the absolute error

introduces biases in the parameter estimates compared to the relative error.
(PNG)

**S5 Fig. Distribution of $L(p)$ with an absolute error when a single observation is randomly sampled at each time point, with the smallest observation (blue) and the largest observation (red) selected at $t = 1$.** The distribution of $L(p)$ varies according to the magnitude of the observations. As the $L(p)$ distribution associated with the larger observation tends to have higher overall values, the likelihood of it being ultimately selected decreases.
(PNG)

**S6 Fig. Estimation of the underlying parameter distribution of the exponential model using EPD with relative error.** (a) When the shape of the underlying distribution is unimodal (True), $L(p)$ with relative error in EPD estimation is not significantly affected by the threshold values (0.1, 0.01, 0.0001). (b) Conversely, if the underlying distribution is bimodal (True), the estimated shape can differ from the underlying distribution depending on the threshold values.
(PNG)

**S7 Fig. We estimate the parameter distributions using RCS data from Fig 3(b-c, left) after removing 90% of the observation data points in three scenarios: 1) removal of data at $t = 0.75$, 2) removal of data at $t = 0.5$, and $t = 0.75$, and 3) removal of data at $t = 0.25$, $t = 0.5$, and $t = 0.75$ (Fig 3, left—Dropped points).** (a) When the underlying parameter distribution is bimodal (right, True), EPD accurately estimates the True parameter distribution (EPD 0.75) using RCS data after removing 90% of the observation data points at $t = 0.75$. Similarly, when 90% is removed at $t = 0.5$ and $t = 0.75$, the estimation using EPD remains close to the underlying distribution (EPD 0.5, 0.75). However, when 90% is removed at $t = 0.25$, $t = 0.5$, and $t = 0.75$, a bias is observed in the height of the peak (EPD 0.25, 0.5, 0.75). (b) When the underlying parameter distribution is trimodal (right, True), EPD can estimate the underlying distribution using RCS data after removing 90% of the data points at $t = 0.75$ (right, EPD 0.75). Unlike the bimodal case (a), the peak heights differ from the True distribution in two cases: 1) when observation data points are removed at $t = 0.5$ and $t = 0.75$ (right, EPD 0.5, 0.75), and 2) when points are removed at $t = 0.25$, $t = 0.5$, and $t = 0.75$ (right, EPD 0.25, 0.5, 0.75). Notably, although the missing data yields the bias in the heights, EPD can still estimate the shape of the parameter distribution, stemming from the heterogeneity of the data.
(PNG)

**S1 Algorithm. Approximate Bayesian Computation(ABC).**
(PDF)

**S2 Algorithm. Metropolis-Hastings algorithm for Bayesian Inference.**
(PDF)

## Author Contributions

**Conceptualization:** Hyeontae Jo, Hyung Ju Hwang.

**Data curation:** Sung Woong Cho, Hyung Ju Hwang.

**Formal analysis:** Hyeontae Jo, Sung Woong Cho, Hyung Ju Hwang.

**Funding acquisition:** Hyeontae Jo, Hyung Ju Hwang.

**Investigation:** Sung Woong Cho.

**Methodology:** Hyeontae Jo, Sung Woong Cho.

**Project administration:** Hyung Ju Hwang.

**Software:** Sung Woong Cho.

**Validation:** Hyeontae Jo, Sung Woong Cho.

**Visualization:** Hyeontae Jo, Sung Woong Cho.

**Writing – original draft:** Hyeontae Jo, Sung Woong Cho, Hyung Ju Hwang.

**Writing – review & editing:** Hyeontae Jo, Hyung Ju Hwang.

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
