## [Decision Letter · Decision Letter 0]

20 Aug 2024

Dear Prof. Hwang,

Thank you very much for submitting your manuscript "Estimating the Distribution of Parameters in Differential Equations with Repeated Cross-Sectional Data" for consideration at PLOS Computational Biology.

As with all papers reviewed by the journal, your manuscript was reviewed by members of the editorial board and by several independent reviewers. In light of the reviews (below this email), we would like to invite the resubmission of a significantly-revised version that takes into account the reviewers' comments.

When revising the manuscript, please address the comments of the reviewers, especially those of Reviewer 3.

We cannot make any decision about publication until we have seen the revised manuscript and your response to the reviewers' comments. Your revised manuscript is also likely to be sent to reviewers for further evaluation.

Sincerely,

Peter Kim

Guest Editor

PLOS Computational Biology

Stacey Finley

Section Editor

PLOS Computational Biology

We would like to reconsider this article after major revisions. When revising the manuscript, please address the comments of all the reviewers, especially those of Reviewer 3.

Reviewer's Responses to Questions

**Comments to the Authors:**

Reviewer #1: Review Report on PCOMPBIOL-D-24-00903, “Estimating the Distribution of Parameters in Differential Equations with Repeated Cross-Sectional Data”

The manuscript proposes a novel parameter estimation method for when the given observations are repeated cross-sectional (RCS) data. The problem of model fitting using RCS is prevalent in biomedical research, and hence the manuscript addresses an important methodological problem. The proposed model is presented without a rigorous theoretical justification, but the authors provide a wide range of examples and simulation studies to demonstrate that their methods work well for common mathematical biology problems. I have identified only a few minor issues to raise:

1. The authors could use some copy editing to improve the writing. Currently, there are too many awkward sentences throughout the manuscript.

2. The presented method is very similar to the approximate Bayesian computation (ABC) approach. Proper citations to the literature on ABC would be useful.

3. In Figures 3b and 3c, while the different modes are correctly identified by the proposed algorithm, the weights for the modes differ from the true weights. For example, in Figure 3c, EPD puts more weight on the parameter value of 2 even when there is no noise. Some comments or analysis on why this happens are needed.

4. When discussing GP-based calibration, please include appropriate references such as Kennedy and O’Hagan (2001) in JRSSB.

References

Kennedy, M. C., & O'Hagan, A. (2001). Bayesian calibration of computer models. Journal of the Royal Statistical Society: Series B (Statistical Methodology), 63(3), 425-464.

Reviewer #2: This paper introduces a novel method for parameter estimation in differential equations using repeated cross-sectional (RCS) data. Traditional methods for parameter estimation in differential equations often lead to data information loss, especially when dealing with RCS data. The authors present the Estimation of Parameter Distribution (EPD) method, which aims to accurately capture the distribution of parameters without losing data information. EPD involves generating synthetic time trajectories from RCS data, estimating parameters by minimizing the discrepancy between these trajectories and the true solution of the differential equation, and selecting parameters based on the scale of discrepancy. The authors evaluate EPD's performance across several models and apply it to real-world datasets, showing its advantage in capturing the shape of parameter distributions and addressing heterogeneity within systems.

The introduction of the EPD method seems novel and addresses the limitations of traditional parameter estimation methods. By focusing on RCS data, the authors present a solution that retains data information and estimates parameter distributions. The manuscript evaluates the EPD method across a few models, including exponential growth, logistic population models, and target cell-limited models with delayed virus production. The application of EPD to a few real-world datasets, such as amyloid beta accumulation and viral load, shows the practical utility of the method. The manuscript is also well-organized, and the results are clearly presented.

Several points could be worth including for elaboration and discussion:

1. The EPD method involves generating a large number of synthetic trajectories and estimating parameters for each, which can be computationally intensive. The manuscript does not provide a detailed discussion of the computational cost and potential strategies to mitigate it.

2. The determination of the scaling factor CC in the accept probability function seems critical for the EPD method's accuracy. However, the manuscript did not provide an analysis of how different values of CC impact the results.

3. The authors mention the use of a logistic transformation in the accept probability calculation. The manuscript could benefit from a discussion on alternative transformations and their potential effects on the parameter estimation process.

4. The EPD method shows promise in the models evaluated, but its applicability to other types of differential equations and systems is not explored.

Reviewer #3: This manuscript seeks to address a problem in parameter estimation when working with discrete time series data, where individual measurements are available, but not individual histories. The authors have proposed a novel method to estimate parameters in dynamical systems models of these data. However, I cannot recommend it for publication on this journal, primarily because the method proposed needs to be more fully developed to make a convincing case. I also have several concerns/suggestions for the authors that I hope will be helpful:

1. A major concern is that the authors only compare their method in one case to GP method. However, if the goal is to arrive at probability distributions of model parameters, then the method should also be compared with other Bayesian Estimation methods eg Metropolis Hastings. For which, priors could be multimodal, possibly circumventing the issues faced by GP in capturing true multimodal distributions.

2. The test cases, eg Fig 1, 2, etc are very artificial. I understand that these are demonstrations that the method proposed can successfully recover the true multimodal distributions on model parameters, but equally, a modeler applying classic methods to these data would notice that the data is not from a single but from multimodal distributions on model parameters and presumably account for that before curve fitting.

3. The fits in Figure 4, panel (c) right box show a number of parameter combinations that were estimated but are very far from true parameter estimates. Given that the model is the logistic model, and hence very simple, it is concerning that the proposed method can have a high number of false positives that are such big outliers.

4. The parameter estimates in Fig 8 and 9 are very scattered. Is it reasonable to expect that the biological data being fitted can admit such diverse parameter values? This points to issues of parameter identifiability, and something that a modeler would attempt to address before undertaking the estimation itself. Also, line 171 on Page 11 seems to have a typo, the parameter names do not match those on Figure 9.

5. On line 7, the authors say "data is collected over time measuring the same 7 variables with different samples or populations at each time point", yet in methods, line 220, they say "Each Yi includes J observed data points at time ti". This is confusing and needs to be clarified. Were the number of observations constant at each time point or did these change? If they change, then sampling from the data points at each time ti can introduce a bias becase sampling from fewer observations will over emphasize their importance relative to data at time points with more observations

6. How big should N be (line 226), ie, how many artificial trajectories should be generated from the data? How does this depend on the number of data points, the number of model parameters, the variance in the data etc etc?

7. Equation (3), why this objective function? What if we minimize simply sum of squares? Or employ some other minimization scheme?

8. Is there a theoretical basis for the definition of an (line 249)? The authors themselves point to challenges with selecting 'c' which seems ad hoc at present. Is there any guidance on how to choose c? what about other forms for an?

9. What about estimation multiple data time courses simultaneously? So if a model has 2+ equations an we had measurements of 2+ variables, then how would this method be amended, or how would it perform?

10. The authors also hint at the method being computationally intensive and with the simplicity of the models they considered here, this can be potentially a bottleneck that may be undesirable.

**Have the authors made all data and (if applicable) computational code underlying the findings in their manuscript fully available?**

Reviewer #1: Yes

Reviewer #2: Yes

Reviewer #3: None

PLOS authors have the option to publish the peer review history of their article (what does this mean?). If published, this will include your full peer review and any attached files.

Reviewer #1: No

Reviewer #2: No

Reviewer #3: No
---

## [Decision Letter · Decision Letter 1]

4 Dec 2024

Dear Prof. Hwang,

We are pleased to inform you that your manuscript 'Estimating the Distribution of Parameters in Differential Equations with Repeated Cross-Sectional Data' has been provisionally accepted for publication in PLOS Computational Biology.

Best regards,

Peter Kim

Guest Editor

PLOS Computational Biology

Stacey Finley

Section Editor

PLOS Computational Biology

Feilim Mac Gabhann

Editor-in-Chief

PLOS Computational Biology

Jason Papin

Editor-in-Chief

PLOS Computational Biology

Reviewer's Responses to Questions

**Comments to the Authors:**

Reviewer #2: The authors have addressed my comments.

Reviewer #3: I am satisfied with the revisions

**Have the authors made all data and (if applicable) computational code underlying the findings in their manuscript fully available?**

Reviewer #2: None

Reviewer #3: None

PLOS authors have the option to publish the peer review history of their article (what does this mean?). If published, this will include your full peer review and any attached files.

Reviewer #2: No

Reviewer #3: No

---

## [Editor Report · Acceptance letter]

16 Dec 2024

PCOMPBIOL-D-24-00903R1 

Estimating the Distribution of Parameters in Differential Equations with Repeated Cross-Sectional Data

Dear Dr Hwang,

I am pleased to inform you that your manuscript has been formally accepted for publication in PLOS Computational Biology. Your manuscript is now with our production department and you will be notified of the publication date in due course.

With kind regards,

Dorothy Lannert
